# Design, Synthesis and Biological Evaluation of Pentacyclic Triterpene Derivatives: Optimization of Anti-ABL Kinase Activity

**DOI:** 10.3390/molecules24193535

**Published:** 2019-09-30

**Authors:** Halil I. Ciftci, Mohamed O. Radwan, Safiye E. Ozturk, N. Gokce Ulusoy, Ece Sozer, Doha E. Ellakwa, Zeynep Ocak, Mustafa Can, Taha F.S. Ali, Howaida I. Abd-Alla, Nurettin Yayli, Hiroshi Tateishi, Masami Otsuka, Mikako Fujita

**Affiliations:** 1Department of Drug Discovery, Science Farm Ltd., 1-7-30-805 Kuhonji, Chuo-ku, Kumamoto 862-0976, Japan; hiciftci@kumamoto-u.ac.jp (H.I.C.); mohamedradwan@kumamoto-u.ac.jp (M.O.R.); 2Medicinal and Biological Chemistry Science Farm Joint Research Laboratory, Faculty of Life Sciences, Kumamoto University, 5-1 Oe-honmachi, Chuo-ku, Kumamoto 862-0973, Japan; profdoha@gmail.com (D.E.E.); mustafacan80@yahoo.com (M.C.); htateishi@kumamoto-u.ac.jp (H.T.); 3Chemistry of Natural Compounds Department, Pharmaceutical and Drug Industries Research Division, National Research Centre, Dokki 12622, Cairo, Egypt; howaida_nrc@yahoo.com; 4Chemistry Department, Faculty of Science, Ege University, Erzene Mahallesi, Genclik Caddesi, Bornova/Izmir 35040, Turkey; sfymrt14@gmail.com (S.E.O.); ng.ulusoy@gmail.com (N.G.U.); ecsozer95@gmail.com (E.S.); 5Department of Biochemistry, Faculty of Pharmacy (Girls), Al-Azhar University, Nasr City 11651, Cairo, Egypt; 6Department of Microbiology, Kocaeli State Hospital, Cedit Mahallesi Gunes Cad, Hastane Yolu Sk, Kocaeli 41300, Turkey; zeynepocak@yahoo.com; 7Department of Engineering Sciences, Faculty of Engineering and Architecture, Izmir Katip Celebi University, Havaalani Sosesi Caddesi No:25, Cigli/Izmir 35620, Turkey; 8Medicinal Chemistry Department, Faculty of Pharmacy, Minia University, Minia 61519, Egypt; 9Faculty of Pharmacy, Karadeniz Technical University, Trabzon 61080, Turkey; yayli@ktu.edu.tr

**Keywords:** leukemia, chronic myelogenous leukemia, ABL kinase, pentacyclic triterpenes, gypsogenin, apoptosis

## Abstract

Imatinib, an Abelson (ABL) tyrosine kinase inhibitor, is a lead molecular-targeted drug against chronic myelogenous leukemia (CML). To overcome its resistance and adverse effects, new inhibitors of ABL kinase are needed. Our previous study showed that the benzyl ester of gypsogenin (**1c**), a pentacyclic triterpene, has anti-ABL kinase and a subsequent anti-CML activity. To optimize its activities, benzyl esters of carefully selected triterpenes (**PT1–PT6**), from different classes comprising oleanane, ursane and lupane, and new substituted benzyl esters of gypsogenin (**GP1–GP5**) were synthesized. All of the synthesized compounds were purified and charachterized by different spectroscopic methods. Cytotoxicity of the parent triterpenes and the synthesized compounds against CML cell line K562 was examined; revealing three promising compounds **PT5**, **GP2** and **GP5** (IC_50_ 5.46, 4.78 and 3.19 μM, respectively). These compounds were shown to inhibit extracellular signal-regulated kinase (ERK) downstream signaling, and induce apoptosis in K562 cells. Among them, **PT5** was identified to have in vitro activity (IC_50_ = 1.44 μM) against ABL1 kinase, about sixfold of **1c**, which was justified by molecular docking. The in vitro activities of **GP2** and **GP5** are less than **PT5**, hence they were supposed to possess other more mechanisms of cytotoxicity. In general, our design and derivatizations resulted in enhancing the activity against ABL1 kinase and CML cells.

## 1. Introduction

Leukemia, a group of blood cancer which initiates in the bone marrow resulting in an abnormal production of white blood cells, is the leading cause of cancer-related deaths in a wide population range, from children to aged people. In 2016, the annual number of new cases of leukemia was 467,000 worldwide, with 310,000 leukemia deaths [1]. Based on disease progression and cell types, leukemia is classified into four main types, acute lymphocytic leukemia (ALL), chronic lymphocytic leukemia (CLL), acute myelogenous leukemia (AML) and chronic myelogenous leukemia (CML) [2]. 

CML is a slow-growing type of leukemia characterized by a chromosome translocation between chromosome 9 and 22 to form the Philadelphia (Ph) chromosome [3,4,5,6]. The Ph chromosome contains the connected breakpoint cluster region (BCR) and the Abelson (ABL) oncogene, resulting in BCR–ABL fusing gene with constitutive tyrosine kinase activity [6,7,8,9]. The tyrosine kinase protein plays an important role in the activation of signalling pathways (Janus kinase: JAK, signal transducers and activators of transcription: STAT, Src family kinase: SFK, phosphoinositide 3-kinase: PI3K, protein kinase B: AKT and extracellular signal-regulated kinase: ERK) and cellular processes such as growth, differentiation, and metabolism [10,11,12]. The introduction of imatinib mesylate, a representative molecular-targeted drug and the first tyrosine kinase inhibitor (TKI) with specific activity against ABL, controls CML in chronic phase. However, a significant percentage of CML patients treated with imatinib developed drug resistance and adverse side effects [13,14,15]. Second and third generation of TKIs such as dasatinib, nilotinib and ponatinib have been developed to overcome these major limitations, but patients treated with these TKIs still did not achieve desired CML treatment and some patients were left with unexpected lung and vascular problems [16,17,18,19]. Hence, there is an urgent need for new therapeutic agents for CML patients.

In recent years, natural products and their derivatives have attracted great deal of interest as a source for the development of new diverse bioactive compounds due to their high safety profile and cost effectiveness [20,21,22,23,24,25,26]. Among them, pentacyclic triterpenes are the subjects of numerous studies carried out by chemists engaged in organic synthesis and pharmacology. A large number of pentacyclic triterpenes such as glycyrrhetic acid, oleanolic acid, ursolic acid, betulinic acid and their derivatives have been reported to exhibit various biological activities including antiinflammatory, antimicrobial, antiviral, antioxidant, antitumor and anticancer [27,28,29,30,31,32]. However, a pentacyclic triterpene named gypsogenin has not yet received significant attention in drug discovery. Gypsogenin extracted from *Gypsophila* with sugar chains is a unique pentacyclic triterpene possessing an aldehyde group at its C4 distinguished from other pentacyclic triterpenes. In our previous studies [33,34] and [35], it was demonstrated that gypsogenin and its derivatives exhibit remarkable activities against diverse human cancer cell lines. Furthermore, we recently discovered that the benzyl ester of gypsogenin compound **1c** has anti-ABL kinase and anti-CML activities [36]. To the best of our knowledge, this was the first study to show the inhibitory activity of pentacyclic triterpenoids and their derivatives on ABL kinase. In this study, we conduct biological evaluation of some pentacyclic triterpenes and their derivatives including newly synthesized compounds to optimize anti-ABL kinase and anti-CML activities.

## 2. Results and Discussion

As representative pentacyclic triterpenes, we selected six compounds, asiatic acid (**AA**), betulinic acid (**BA**), glycyrrhetinic acid (**GA**), hederagenin (**HE**), oleanolic acid (**OA**) and ursolic acid (**UA**) (Figure 1). **AA** and **UA** represent the ursane type, and **BA** represents the lupane type. **GA**, **HE** and **OA** are of the oleanane type. On these pentacyclic triterpenes and each benzyl ester like **1c** (**PT1**–**PT6**), biological activities were examined. Furthermore, five new substituted benzyl esters of gypsogenin **GP1**–**GP5** were also designed. **GP1** has a 4-methoxy substitution as an electron-donating group whereas other compounds have electron-withdrawing substations such as fluoro-, trifluoromethyl-, and cyano- groups.

The synthesis of benzyl esters of different triterpenes **PT1**–**PT6** and the new substituted benzyl esters of gypsogenin **GP1**–**GP5** was performed as outlined in Scheme 1. **AA**, **BA**, **GA**, **HE**, **OA** and **UA** were reacted with benzyl bromide in the presence of potassium carbonate as a base to afford the corresponding esters **PT1**, **PT2**, **PT3**, **PT4**, **PT5**, and **PT6** respectively in good yields (44–81%). In a similar way, **GP** was reacted with different substituted benzyl bromides producing five new compounds **GP1**, **GP2**, **GP3**, **GP4**, and **GP5** in 40–48% yield. 

To evaluate the antiproliferative effects of all compounds against CML cell line K562, MTT assay was carried out. The results were summarized in Table 1. Imatinib was used as a control. IC_50_ values of free triterpenes were detected to be higher than 10 µM. Introduction of benzyl ester group into triterpene led to an increase in anticancer activity on K562 except for **PT3** and **PT6**, based on **GA** and **UA**, respectively. Specially, **PT4** has more than 15 times stronger cytotoxicity than its parent triterpene **HE**. The IC_50_ values of all new gypsogenin derivatives **GP1**–**GP5** were lower than 6 µM. In order to investigate whether **PT1**–**PT6** and **GP1**–**GP5** are toxic to healthy blood cells or not, the cytotoxic effects of these compounds on peripheral blood mononuclear cells (PBMC) were also examined. Each compound, except **HE** which lacks cytotoxicity against K562, showed a considerable selectivity index (SI). Notably, free triterpenes **GP** and **OA** have IC_50_ of around 13 μM against K562 and SI of more than 7. We identified **GP2**, **GP5** and **PT5** as the strongest and selective anticancer agents on CML cell line. These three compounds were also more selective than imatinib.

The cytotoxic effects of these compounds were also evaluated on different leukemic cells (HL-60, MT-2 and Jurkat), HeLa, MCF7 and A549 cell lines as shown in Table 2. Among them, the cyano-substituted compound **GP5** were found as the most effective on all cancer cell lines. **GP2** and **PT5** were inactive against A549 cell line at 100 µM concentration. **PT5** was less active compared to **GP2** and **GP5** on HL-60, Jurkat, HeLa and MCF7 cell lines. **PT5** exhibited selectivity against leukemia cells like imatinib.

Due to their significant anticancer potentials on CML, **GP2**, **GP5** and **PT5** were further investigated for their apoptotic activities in K562 cells using Hoechst 33342/annexin V/ethidium homodimer III staining method. K562 cells were treated with **GP2**, **GP5** and **PT5** at their IC_50_ concentrations for 8 h, then stained and observed by a florescence microscope (Figure 2a). The apoptotic effects of **GP2**, **GP5**, **PT5** and imatinib on CML cell line were determined as 24%, 9.5%, 14% and 18%, respectively, while their late apoptotic/necrotic effects were determined as 4.5%, 5.1%, 11% and 5.6% respectively (Figure 2b). According to these findings, **GP2** having 3,5-ditrifluoromethyl substitutions induced more pronounced apoptosis of K562 cells than **GP5**, **PT5** and imatinib. 

As a consequence of the central role of ABL kinase in cellular processes including cell replication and apoptosis, **GP2**, **GP5** and **PT5** were selected for further investigation of the potential inhibitory activity against ABL kinase using multipoint dose-response experiments (Table 3). Among them, **PT5** was found to be the most potent ABL kinase inhibitor with the IC_50_ value of 1.44 µM, followed by **GP5** and **GP2** with IC_50_ values of 6.16 µM and 7.19 µM, respectively. IC_50_ of **PT5** is six times stronger than **1c**. Although it is still incomparable to imatinib (IC_50_ 0.21 µM), they showed similar cellular activity (IC_50_ ~5.5 μM) against K562 cells (Table 1) and similar selectivity against various cancer cell lines (Table 2). **PT5** is considered to express more efficient cellular activity, but the detailed reason remains elusive. 

To investigate this outstanding activity of **PT5**, we conducted a molecular docking study into the ATP-binding site of ABL kinase. Herein, we used the most updated version of Molecular Operating Environment (MOE) software that explains the slightly shifted placement of **1c** from our previous work [36]. Both compounds demonstrated a similar and superimposed binding mode typically settling three CH-π interaction with His361, Asp381 and Phe401. Their hydrophobic fused cyclohexyl rings lie in proximity to hydrophobic residues Val289, Phe359 and Ile360, Ala380 and Phe382 (Figure 3). We believe that the presence of the polar aldehyde group on C4 of compound **1c** (binding energy -7.11) in proximity to the hydrophobic residues Ala399 and Phe401 is unfavorable compared to the lower methyl group of **PT5** which is at the same position (binding energy -7.31). Finally, the rigid skeleton of **PT5** limits its deep burial into the pocket as fulfilled by imatinib.

In order to explore the inhibitory potency of **GP2, GP5,** and **PT5** on downstream signaling of BCR-ABL kinase such as rapidly accelerated fibrosarcoma (Raf)/mitogen-activated protein kinase/extracellular signal-regulated kinase (MEK)/extracellular signal-regulated kinase (ERK), ERK1/2 and their phosphorylated forms in K562-treated with **GP2, GP5,** or **PT5** were examined at 10 μM concentration for 6 h (Figure 4). In the presence of the titled compounds, amount of ERK1/2 moderately decreased. In contrast, the reduction of phosphorylated ERK1/2 was prominent where the three compounds exhibited inhibitory effect of ERK phosphorylation in a similar pattern to imatinib. The inhibitory activity against ERK phoshorylation is not in accordance with cytotoxicity data in K562 cells (Table 1). This suggests that there is a pathway other than ERK signaling suppressed by the titled compounds.

## 3. Materials and Methods 

### 3.1. Chemistry

All reactions were performed in an efficient fume hood. Chemicals were purchased from Sigma-Aldrich (St. Louis, MO, USA), Honeywell Fluka (Morristown, NJ, USA), Kanto Chemical (Tokyo, Japan), Nacalai Tesque (Kyoto, Japan), Tokyo Chemical Industry (Tokyo, Japan) and FUJIFILM Wako (Osaka, Japan). Commercially available reagent-grade chemicals were used without further purification. The reaction progress was monitored by a thin layer chromatography (TLC) on precoated plates [Merck (Darmstadt, Germany) TLC sheets silica 60 F254]. The flash column chromatography was carried out on Silica Gel 60N [40–100 mesh, Kanto Chemical (Tokyo, Japan)]. Melting points were determined on a Yanaco (Kyoto, Japan) melting point apparatus and were uncorrected. ^1^H and ^13^C NMR spectra were registered on a Bruker (Billerica, MA, USA) Avance 600 (600 MHz). Chemical shits are referenced to TMS. Mass spectra (MS) and high-resolution mass spectra (HRMS) were recorded on a JEOL (Tokyo, Japan) JMS-DX303HF using positive fast atom bombardment (FAB) with 3-nitrobenzyl alcohol as the matrix. 

### 3.2. General Method of Preparation of **PT1**–**PT5** and **GP1**–**GP5** Compounds

A 0.5 mmol of a triterpenic compound was dissolved in 5 mL dry DMF followed by the addition of anhydrous K_2_CO_3_ (1.5 mmol) and stirred for 30 min. The appropriate free or substituted benzyl bromide was added and the mixture was stirred for an additional 6 h at room temperature. Upon the consumption of the reactant, 5 mL of the brine solution was added and the mixture was extracted with ethyl acetate. The organic phase was washed with water until pH = 7, dried over anhydrous sodium sulfate, filtered and concentrated under vacuo. The resulted residue was applied for column chromatography using an appropriate eluent [31].

Asiatic acid benzyl ester (**PT1**) [37]. Compound **PT1** was obtained from **AA** and benzyl bromide as a white solid (173 mg, 60%). ^1^H NMR (600 MHz, CDCl_3_) δ: 7.36–7.30 (m, 5H, *H*-Ar), 5.23 (t, *J* = 3.6 Hz, 1H), 5.09 (d, *J* = 12.5 Hz, 1H, Bn-CH*H’*), 4.98 (d, *J* = 12.5 Hz, 1H, Bn-C*H*H’), 3.77–3.73 (m, 1H), 3.64 (d, *J* = 10.6 Hz, 1H), 3.51 (brs, 1H), 3.41 (d, *J* = 9.8 Hz, 2H), 2.64 (brs, 2H), 2.26 (d, *J* = 11.3 Hz, 1H), 2.01–1.86 (m, 4H), 1.80–1.67 (m, 5H), 1.63–1.55 (m, 2H), 1.50–1.45 (m, 2H), 1.36–1.25 (m, 5H), 1.07 (s, 3H), 1.05–1.02 (m, 2H), 1.00 (s, 3H), 0.93 (d, *J* = 6.5 Hz, 3H), 0.86 (s, 3H), 0.85 (d, *J* = 6.5 Hz, 3H), 0.63 (s, 3H). ^13^C NMR (150 MHz, CDCl_3_) δ: 177.29, 138.22, 136.36, 128.42, 128.16, 127.97, 125.41, 80.56, 70.62, 68.77, 66.01, 52.87, 49.15, 48.12, 47.48, 46.29, 42.44, 42.14, 39.56, 39.11, 38.84, 38.15, 36.62, 32.68, 30.66, 27.92, 24.23, 23.64, 23.33, 21.17, 18.33, 17.16, 17.05, 17.01, 12.79. MS (FAB) *m*/*z* 601.2 (M + Na)^+^; HRMS (FAB). Calcd for C_37_H_54_O_5_Na: 601.3869. Found: 601.3868. (Appendix A).

Betulinic acid benzyl ester (**PT2**) [38]. Compound **PT2** was obtained from **BA** and benzyl bromide as a white solid (136 mg, 50%). ^1^H NMR (600 MHz, CDCl_3_) δ: 7.35–7.29 (m, 5H, *H*-Ar), 5.12 (d, *J* = 12.3 Hz, 1H, Bn-CH*H’*), 5.07 (d, *J* = 12.3 Hz, 1H, Bn-C*H*H’), 4.70 (d, *J* = 2.3 Hz, 1H), 4.57 (dd, *J* = 2.3, 1.4 Hz, 1H), 3.17–3.13 (m, 1H), 3.02–2.98 (m, 1H), 2.27–2.24 (m, 1H), 2.18–2.14 (m, 1H), 1.89–1.83 (m, 2H), 1.63–1.56 (m, 3H), 1.54 (s, 3H), 1.52–1.16 (m, 14H), 1.08 (dt, *J* = 13.5, 3.1 Hz, 1H), 0.99 (td, *J* = 12.8, 4.6 Hz, 1H), 0.93 (s, 3H), 0.92 (s, 3H), 0.86 (td, *J* = 12.8, 3.9 Hz, 1H), 0.78 (s, 3H), 0.74 (s, 3H), 0.73 (s, 3H), 0.65–0.63 (m, 1H). ^13^C NMR (150 MHz, CDCl_3_) δ 175.82, 150.59, 136.52, 128.50, 128.25, 128.06, 109.57, 79.01, 65.74, 56.58, 55.39, 50.60, 49.50, 46.96, 42.42, 40.69, 38.88, 38.75, 38.24, 37.21, 36.96, 34.34, 32.15, 30.62, 29.60, 28.00, 27.45, 25.58, 20.91, 19.39, 18.32, 16.13, 15.86, 15.36, 14.69. MS (FAB) *m*/*z* 569.5 (M + Na)^+^; HRMS (FAB). Calcd for C_37_H_54_O_3_Na: 569.3971. Found: 569.4046. (Appendix A).

Glycyrrhetic acid benzyl ester (**PT3**) [31,38]. Compound **PT3** was obtained from **GA** and benzyl bromide as a white solid (123 mg, 44%). ^1^H NMR (600 MHz, CDCl_3_) δ: 7.40–7.32 (m, 5H, *H*-Ar), 5.55 (s, 1H), 5.20 (d, *J* = 12.2 Hz, 1H, Bn-CH*H’*), 5.09 (d, *J* = 12.2 Hz, 1H, Bn-C*H*H’), 3.24–3.21 (m, 1H), 2.79 (dt, *J* = 13.4, 3.5 Hz, 1H), 2.32 (s, 1H), 2.05–2.01 (m, 3H), 1.93 (ddd, *J* = 13.6, 4.3, 2.7 Hz, 1H), 1.81 (td, *J* = 13.6, 4.3 Hz, 1H), 1.67–1.61 (m, 4H), 1.59–1.57 (m, 1H), 1.47–1.42 (m,1H), 1.39 (ddd, *J* = 11.0, 6.9, 2.7 Hz, 2H), 1.35 (s, 3H), 1.32–1.26 (m, 4H), 1.16 (s, 3H), 1.13 (s, 3H), 1.11 (s, 3H), 1.00 (s, 3H), 0.99–0.94 (m, 1H), 0.80 (s, 3H), 0.73 (s, 3H), 0.69 (dd, *J* = 11.9, 1.5 Hz, 1H). ^13^C NMR (150 MHz, CDCl_3_) δ: 200.11, 176.21, 168.97, 136.16, 128.63, 128.56, 128.31, 128.26, 78.80, 66.24, 61.82, 54.98, 48.23, 45.37, 44.00, 43.20, 41.12, 39.18, 39.15, 37.68, 37.11, 32.80, 31.80, 31.20, 28.43, 28.30, 28.12, 27.34, 26.50, 26.44, 23.37, 18.70, 17.52, 16.38, 15.58. MS (FAB) *m*/*z* 561.5(M + H)^+^; HRMS (FAB). Calcd for C_37_H_53_O_4_: 561.3944. Found: 561.3970. (Appendix A).

Hederagenin benzyl ester (**PT4**) [39]. Compound **PT4** was obtained from **HE** and benzyl bromide as a white solid (160 mg, 57%). ^1^H NMR (600 MHz, CDCl_3_) δ: 7.36–7.29 (m, 5H, *H*-Ar), 5.28 (t, *J* = 3.6 Hz, 1H), 5.09 (d, *J* = 12.6 Hz, 1H, Bn-CH*H’*), 5.04 (d, *J* = 12.6 Hz, 1H, Bn-C*H*H’), 3.72 (d, *J* = 10.3 Hz, 1H), 3.64–3.62 (m, 1H), 3.42 (d, *J* = 10.3 Hz, 1H), 2.90 (dd, *J* = 13.7, 4.3 Hz, 1H), 2.55 (brs, 1H), 2.34 (brs, 1H), 1.97 (td, *J* = 13.9, 4.4 Hz, 1H), 1.85 (dd, *J* = 8.9, 3.6 Hz, 2H), 1.71 (td, *J* = 13.9, 4.4 Hz, 1H), 1.67–1.62 (m, 3H), 1.57–1.51 (m, 2H), 1.40–1.33 (m, 3H), 1.42–1.33 (m, 4H), 1.23–1.13 (m, 3H), 1.12 (s, 3H), 1.03 (dt, *J* = 13.5, 3.1 Hz, 1H), 0.98–0.94 (m, 1H), 0.93 (s, 3H), 0.92 (s, 3H), 0.89 (s, 3H), 0.89 (s, 3H), 0.84 (dd, *J* = 10.6, 3.1 Hz, 1H), 0.60 (s, 3H). ^13^C NMR (150 MHz, CDCl_3_) δ: 177.46, 143.69, 136.45, 128.42, 128.00, 127.92, 122.44, 76.89, 72.16, 65.95, 49.82, 47.64, 46.76, 45.90, 41.85, 41.74, 41.41, 39.31, 38.14, 36.92, 33.88, 33.10, 32.54, 32.39, 30.71, 27.64, 26.84, 25.90, 23.65, 23.39, 23.07, 18.50, 16.92, 15.67, 11.37. MS (FAB) *m*/*z* 585.2 (M + Na)^+^; HRMS (FAB). Calcd for C_37_H_54_O_4_Na: 585.3920. Found: 585.3912. (Appendix A).

Oleanolic acid benzyl ester (**PT5**) [31,38]. Compound **PT5** was obtained from **OA** and benzyl bromide as a white solid (185 mg, 68%). ^1^H NMR (600 MHz, CDCl_3_) δ: 7.35–7.29 (m, 5H, *H*-Ar), 5.29 (t, *J* = 3.6 Hz, 1H), 5.09 (d, *J* = 12.6 Hz, 1H, Bn-CH*H’*), 5.05 (d, *J* = 12.6 Hz, 1H, Bn-C*H*H’), 3.20 (dt, *J* = 10.1, 4.9 Hz, 1H), 2.90 (dd, *J* = 13.8, 4.4 Hz, 1H), 1.98 (td, *J* = 13.8, 4.1 Hz, 1H), 1.85 (dd, *J* = 8.9, 3.7 Hz, 2H), 1.74–1.59 (m, 7H), 1.57–1.50 (m, 4H), 1.42 (td, *J* = 12.6, 3.7 Hz, 1H), 1.37–1.16 (m, 7H), 1.12 (s, 3H), 1.04 (dt, *J* = 6.6, 3.5 Hz, 1H), 0.98 (s, 3H), 0.92 (s, 3H), 0.90 (s, 3H), 0.88 (s, 3H), 0.77 (s, 3H), 0.71 (dd, *J* = 11.7, 1.8 Hz, 1H), 0.61 (s, 3H). ^13^C NMR (150 MHz, CDCl_3_) δ: 177.46, 143.72, 136.48, 128.42, 127.99, 127.91, 122.53, 79.04, 65.94, 55.25, 47.65, 46.78, 45.92, 41.73, 41.42, 39.33, 38.77, 38.48, 37.04, 33.90, 33.11, 32.76, 32.41, 31.59, 30.71, 28.12, 27.66, 27.23, 25.89, 23.66, 23.42, 23.09, 22.66, 18.34, 16.91, 15.58, 15.31, 14.11. MS (FAB) *m*/*z* 569.4 (M + Na)^+^; HRMS (FAB). Calcd for C_37_H_54_O_3_Na: 569.3967. Found: 569.4046. (Appendix A).

Ursolic acid benzyl ester (**PT6**) [38]. Compound **PT6** was obtained from **UA** and benzyl bromide as a white solid (221 mg, 81%). ^1^H NMR (600 MHz, CDCl_3_) δ: 7.36–7.30 (m, 5H, *H*-Ar), 5.23 (t, *J* = 3.6 Hz, 1H), 5.10 (d, *J* = 12.5 Hz, 1H, Bn-CH*H’*), 4.98 (d, *J* = 12.5 Hz, 1H, Bn-C*H*H’), 3.21 (dd, *J* = 11.3, 4.7 Hz, 1H), 2.26 (dd, *J* = 11.3, 0.9 Hz, 1H), 2.01 (td, *J* = 13.4, 4.5 Hz, 1H), 1.90–1.83 (m, 2H), 1.82–1.76 (m, 1H), 1.73–1.70 (m, 2H), 1.67–1.44 (m, 9H), 1.36–1.25 (m, 6H), 1.07 (s, 3H), 1.05–1.01 (m, 1H), 0.98 (s, 3H), 0.93 (d, *J* = 6.4 Hz, 3H), 0.89 (s, 3H), 0.85 (d, *J* = 6.4 Hz, 3H), 0.77 (s, 3H), 0.70 (dd, *J* = 11.7, 1.3 Hz, 1H), 0.64 (s, 3H). ^13^C NMR (150 MHz, CDCl_3_) δ: 177.29, 138.13, 136.41, 128.41, 128.16, 127.95, 125.74, 79.04, 65.99, 55.26, 52.93, 48.15, 47.60, 42.07, 39.56, 39.12, 38.87, 38.77, 38.67, 36.98, 36.66, 33.07, 31.60, 30.70, 28.17, 27.99, 27.26, 24.28, 23.59, 23.29, 22.66, 21.19, 18.33, 17.04, 17.01, 15.64, 15.46, 14.13. MS (FAB) *m*/*z* 569.3 (M + Na)^+^; HRMS (FAB). Calcd for C_37_H_54_O_3_Na: 569.3971. Found: 569.3970. (Appendix A).

Gypsogenin 4-methoxybenzyl ester (**GP1**). Compound **GP1** was obtained from **GP** and 4-methoxybenzyl bromide as a white solid (123 mg, 42%); mp 78–80 °C. ^1^H NMR (600 MHz, CDCl_3_) δ: 9.39 (s, 1H), 7.26 (d, *J* = 8.8 Hz, 2H, *H*-Ar), 6.86 (d, *J* = 8.8 Hz, 2H, *H*-Ar), 5.28 (t, *J* = 3.6 Hz, 1H), 5.02 (d, *J* = 12.1 Hz, 1H, Bn-CH*H’*), 4.97 (d, *J* = 12.1 Hz, 1H, Bn-C*H*H’), 3.80 (s, 3H), 3.76 (dd, *J* = 11.6, 4.3 Hz, 1H), 2.88 (dd, *J* = 13.9, 4.3 Hz, 1H), 1.95 (td, *J* = 13.6, 4.1 Hz, 1H), 1.87 (dd, *J* = 8.9, 3.6 Hz, 2H), 1.71–1.44 (m, 14H), 1.32 (td, *J* = 13.9, 4.3 Hz, 1H), 1.27–1.25 (m, 1H), 1.19–1.14 (m, 2H), 1.13 (s, 3H), 1.06 (s, 3H), 1.04–1.02 (m, 1H), 1.04–0.97 (m, 3H), 0.93 (s, 3H), 0.91 (s, 3H), 0.89 (s, 3H), 0.59 (s, 3H). ^13^C NMR (150 MHz, CDCl_3_) δ: 207.08, 177.41, 159.46, 143.84, 129.84, 128.59, 122.06, 113.81, 71.93, 65.75, 55.29, 55.19, 48.25, 47.56, 46.65, 45.91, 41.79, 41.42, 39.67, 38.07, 36.01, 33.87, 33.10, 32.31, 32.14, 30.71, 27.60, 26.13, 25.89, 23.65, 23.37, 22.99, 20.76, 16.91, 15.57, 8.95. MS (FAB) *m*/*z* 613.5 (M + Na)^+^; HRMS (FAB). Calcd for C_38_H_54_O_5_Na: 613.3869. Found: 613.3865. (Appendix A).

Gypsogenin 3,5-bis(trifluromethyl)benzyl ester (**GP2**): Compound **GP2** was obtained from **GP** and 3,5-bis(trifluoromethyl)benzyl bromide as a white solid (149 mg, 43%); mp 85–87 °C. ^1^H NMR (600 MHz, CDCl_3_) δ: 9.39 (s, 1H), 7.83 (s, 1H, *H*-Ar), 7.81 (s, 2H, *H*-Ar), 5.28 (t, *J* = 3.6 Hz, 1H), 5.20 (d, *J* = 13.2 Hz, 1H, Bn-CH*H’*), 5.13 (d, *J* = 13.2 Hz, 1H, Bn-C*H*H’), 3.76 (dd, *J* = 11.5, 4.5 Hz, 1H), 2.89 (dd, *J* = 13.7, 4.1 Hz, 1H), 2.01 (td, *J* = 13.7, 4.1 Hz, 1H), 1.87–1.84 (m, 2H), 1.74–1.44 (m, 14H), 1.36 (td, *J* = 13.7, 4.1 Hz, 1H), 1.26–1.17 (m, 3H), 1.15 (s, 3H), 1.06 (s, 3H), 0.98–0.95 (m, 1H), 0.93 (s, 3H), 0.91 (s, 3H), 0.89 (s, 3H), 0.48 (s, 3H). ^13^C NMR (150 MHz, CDCl_3_) δ: 207.10, 177.18, 143.69, 138.97, 131.92 (q, *J* = 33.5 Hz), 128.09, 124.09, 122.32, 122.01–121.91 (m), 71.94, 64.35, 55.15, 48.20, 47.46, 46.88, 45.85, 41.77, 41.38, 39.61, 38.01, 35.97, 33.78, 33.01, 32.39, 31.98, 31.58, 30.66, 27.49, 26.10, 25.89, 23.51, 23.19, 23.11, 22.64, 20.65, 16.67, 15.36, 14.09, 8.95. MS (FAB) *m*/*z* 719.4 (M + Na)^+^; HRMS (FAB). Calcd for C_39_H_50_O_4_F_6_Na: 719.3511. Found: 719.3511. (Appendix A).

Gypsogenin 4-trifluoromethyl benzyl ester (**GP3**). Compound **GP3** was obtained from **GP** and 4-trifluoromethyl benzyl bromide as a white solid (128 mg, 41%); mp 175–177 °C. ^1^H NMR (600 MHz, CDCl_3_) δ: 9.39 (s, 1H), 7.60 (d, *J* = 8.0 Hz, 2H, *H*-Ar), 7.46 (d, *J* = 8.0 Hz, 2H, *H*-Ar), 5.28 (t, *J* = 3.6 Hz, 1H), 5.15 (d, *J* = 12.9 Hz, 1H, Bn-CH*H’*), 5.07 (d, *J* = 12.9 Hz, 1H, Bn-C*H*H’), 3.76 (dd, *J* = 11.4, 4.4 Hz, 1H), 2.89 (dd, *J* = 13.5, 4.1 Hz, 1H), 1.99 (td, *J* = 13.5, 4.1 Hz, 1H), 1.87–1.84 (m, 2H), 1.74–1.19 (m, 7H), 1.55 (ddd, *J* = 18.5, 11.1, 4.1 Hz, 2H), 1.49–1.42 (m, 3H), 1.34 (td, *J* = 13.7, 4.1 Hz, 1H), 1.21 (ddd, *J* = 13.5, 4.1, 2.1 Hz, 2H), 1.14 (s, 3H), 1.06 (s, 3H), 1.04–0.96 (m, 3H), 0.93 (s, 3H), 0.91 (s, 6H), 0.89–0.87 (m, 1H), 0.86–0.83 (m, 1H), 0.50 (s, 3H). ^13^C NMR (150 MHz, CDCl_3_) δ: 207.09, 177.22, 143.73, 140.35, 128.27, 125.42 (q, *J* = 3.8 Hz), 123.15, 122.22, 71.95, 65.07, 55.16, 48.24, 47.50, 46.78, 45.86, 41.78, 41.45, 39.64, 38.06, 35.99, 33.82, 33.06, 32.38, 32.10, 30.70, 29.71, 27.57, 26.12, 25.90, 23.62, 23.35, 23.06, 20.67, 16.77, 15.49, 8.95. MS (FAB) *m*/*z* 651.5 (M + Na)^+^; HRMS (FAB). Calcd for C_38_H_51_O_4_F_3_Na: 651.3637. Found: 651.3605. (Appendix A).

Gypsogenin 3,5-difluorobenzyl ester (**GP4**). Compound **GP4** was obtained from **GP** and 3,5-difluorobenzyl bromide as a white solid (119 mg, 40%); mp 58–60 **°C**. ^1^H NMR (600 MHz, CDCl_3_) δ: 9.40 (s, 1H), 6.86 (dd, *J* = 7.9, 1.8 Hz, 2H, *H*-Ar), 6.76–6.72 (m, 1H, *H*-Ar), 5.32 (t, *J* = 3.6 Hz, 1H), 5.09 (d, *J* = 13.1 Hz, 1H, Bn-CH*H’*), 4.96 (d, *J* = 13.1 Hz, 1H, Bn-C*H*H’), 3.77 (d, *J* = 10.0 Hz, 1H), 2.90 (dd, *J* = 13.5, 4.1 Hz, 1H), 2.00 (td, *J* = 13.7, 4.1 Hz, 1H), 1.89 (dd, *J* = 8.9, 3.6 Hz, 2H), 1.72–1.61 (m, 7H), 1.59 (s, 3H), 1.56–1.44 (m, 4H), 1.38–1.18 (m, 6H),1.15 (s, 3H), 1.06 (s, 3H), 1.05–1.04 (m, 1H), 1.00–0.97 (m, 1H), 0.93 (s, 3H), 0.91 (s, 3H), 0.88 (t, *J* = 7.0 Hz, 1H), 0.59 (s, 3H). ^13^C NMR (150 MHz, CDCl_3_) δ: 207.09, 177.16, 163.86 (d, *J* = 12.5 Hz), 162.21 (d, *J* = 12.5 Hz), 143.72, 140.26, 122.35, 110.55, 110.42, 103.29, 71.94, 64.55, 55.49, 55.17, 48.26, 47.51, 46.83, 45.86, 41.81, 41.45, 39.68, 38.05, 36.02, 33.81, 33.06, 32.39, 32.14, 31.59, 30.70, 27.56, 26.12, 25.91, 23.61, 23.30, 23.07, 22.65, 20.74, 16.85, 15.53, 14.11, 8.96. MS (FAB) *m*/*z* 619.2 (M + Na)^+^; HRMS (FAB). Calcd for C_3__7_H_5__0_O_4_F_2_Na: 619.3575. Found: 619.3576. (Appendix A).

Gypsogenin 4-cyanobenzyl ester (**GP5**). Compound **GP5** was obtained from **GP** and 4-cyanobenzyl bromide as a white solid (140 mg, 48%); mp 175–177 °C. ^1^H NMR (600 MHz, CDCl_3_) δ 9.39 (s, 1H), 7.65 (d, *J* = 8.3 Hz, 2H, *H*-Ar), 7.45 (d, *J* = 8.3 Hz, 2H, *H*-Ar), 5.29 (t, *J* = 3.5 Hz, 1H), 5.16 (d, *J* = 13.3 Hz, 1H, Bn-CH*H’*), 5.05 (d, *J* = 13.3 Hz, 1H, Bn-C*H*H’), 3.76 (dd, *J* = 11.4, 4.1 Hz, 1H), 2.89 (dd, *J* = 13.8, 4.1 Hz, 1H), 2.00 (td, *J* = 13.7, 4.1 Hz, 1H), 1.90–1.77 (m, 2H), 1.73–1.45 (m, 12H), 1.35 (td, *J* = 13.7, 4.1 Hz, 1H), 1.27–1.17 (m, 4H), 1.15 (s, 3H), 1.06 (s, 3H), 1.05–1.04 (m, 1H), 1.00–0.97 (m, 1H), 0.93 (s, 3H), 0.92 (s, 3H), 0.91 (s, 3H), 0.88 (t, *J* = 7.0 Hz, 1H), 0.56 (s, 3H). ^13^C NMR (150 MHz, CDCl_3_) δ 207.12, 177.16, 143.67, 141.70, 132.30, 128.37, 122.27, 118.56, 111.87, 71.95, 64.86, 55.17, 48.20, 47.48, 46.85, 46.84, 45.81, 41.80, 41.79, 41.47, 39.66, 38.05, 35.99, 33.79, 33.04, 32.38, 32.11, 31.58, 30.70, 30.69, 30.68, 30.68, 27.57, 26.11, 25.90, 23.61, 23.35, 23.06, 22.65, 20.71, 16.87, 15.57, 14.11, 8.98. MS (FAB) *m*/*z* 608.5 (M + Na)^+^; HRMS (FAB). Calcd for C_38_H_51_NO_4_Na: 608.3716. Found: 608.3710. (Appendix A).

### 3.3. Biochemistry

#### 3.3.1. Cell Cultures

A549 human lung carcinoma and HeLa human cervical carcinoma cell lines were cultured in Dulbecco’s Modified Eagle Medium (DMEM). MCF7 human breast cancer and leukemic (K562, HL-60, MT-2 and Jurkat) cells, and PBMC (Precision Bioservices, Frederic, MD, USA) were cultured in RPMI 1640. All media were purchased from FUJIFILM Wako (Osaka, Japan) and supplemented with 10% fetal bovine serum (FBS) (Sigma-Aldrich, St. Louis, MO, USA) and 89 μg/mL streptomycin (Meiji Seika Pharma, Tokyo, Japan) at 37 °C in a humid atmosphere and 5% CO_2_. In experiments, all cancer cell lines and PBMC were cultured in 24-well and 96-well plates (Iwaki brand Asahi Glass, Chiba, Japan) at 2 × 10^4^ cells/mL and 1 × 10^6^ cells/mL concentrations, respectively for 24 h. The stock solution of compounds and imatinib in concentrations between 0,1-10 mM were prepared in DMSO (FUJIFILM Wako, Osaka, Japan) and further diluted with fresh culture medium. The concentration of DMSO in the final culture medium was 1%, which had no effect on the cell viability [40].

#### 3.3.2. Cell Viability Assay

The effect of PTs, their derivatives and imatinib on cell viability was assessed by using MTT (Dojindo, Kumamoto, Japan) as previously described in the literature [41,42]. The cells were exposed to various concentrations (1–100 μM) of the compounds for 24 h at 37 °C. The cells were then stained with MTT solution and incubated for additional 4 h. At the end of this period, supernatants were removed and 100 μL DMSO was added to each well to solubilize the formazan crystals. The absorbance of the solution was determined on the Infinitive M1000 (Tecan, Mannedorf, Switzerland) plate reader at a wavelength of 550 nm with background subtraction at 630 nm. All experiments were performed in triplicate and IC_50_ values defined as the drug concentrations that reduced absorbance to 50% of control values were calculated from MTT results. 

#### 3.3.3. Cell Death Detection 

After the cells were incubated with the selected compounds in this series at IC_50_ concentration for 6 h, the apoptotic/necrotic/healthy/detection kit (PromoKine, Heidelberg, Germany) was performed according to the manufacturer`s instructions. The cells were then briefly washed twice with 1 × binding buffer. A staining solution containing 50 μL of 1 × binding buffer and 5 μL of FITC-Annexin V solution, 5 μL of ethidium homodimer III solution, or 5 μL of Hoechst 33342 solution was added, and incubated for 15 min at room temperature in the dark. Then, the cells were washed with 1 × binding buffer, and analyzed by the all-in-one fluorescence microscope Biorevo Fluorescence BZ-9000 (Keyence, Osaka, Japan). The number of healthy cells (Hoechst 33342), apoptotic cells (Annexin V), late apoptotic or necrotic cells (Annexin V and Ethidium homodimer III) and necrotic cells (Ethidium homodimer III) were counted as previously described [43].

#### 3.3.4. Determination of ABL1 Kinase Inhibition

The ABL1 kinase profiling assay protocol was applied according to the manufacturer`s instructions (Promega, Madison, WI, USA) with the modification proposed in [36]. In this system, the ABL1 kinase and its substrate were diluted with 95 µL 2.5x Kinase Buffer and 15 µL of 100 µM ATP solutions, respectively. Then, the reaction of ABL1 kinase was performed using 2 µL of the compound solution at varying concentrations (0.1–30 µM), 4 µL of kinase working stock, and 4 µL of ATP/substrate working stock in the 384-well plate. After 1 h of incubation at room temperature, the activity of the ABL1 kinase was detected using the ADP-Glo Kinase Assay (Promega, Madison, WI, USA) and the inhibitory kinase activity of compounds in dose-response mode was measured by the Infinitive M1000 luminescence plate reader (Tecan, Groding, Austria). The IC_50_ values of tested compounds required to decrease the kinase activity by 50% were calculated. 

#### 3.3.5. Immunoblot Analysis

The K562 cells were cultured with compounds **GP2**, **GP5**, **PT5** and imatinib at 10 µM concentrations. After 6 h of treatment at 37 °C, the cells were lysed in PBS-Laemmli sample buffer and then, immunoblot analysis was conducted using phospho-specific-p44/42 MAPK (Erk1/2) (Thr202/Tyr204) (D13.14.4E) XP Rabbit mAb (1:1000) (Cell Signaling Technology, Danvers, MA, USA) or anti-β-actin clone AC-15 (Sigma-Aldrich, St. Louis, MO, USA). For immunoreactivity detection, the chemiluminescence method was performed as previously described [44,45]. The band intensity was measured using the ImageJ software (NIH, Bethesda, MD, USA). 

### 3.4. Molecular Docking 

Crystal structure of the ABL kinase in complex with imatinib was obtained from Protein Data Bank (PDB: 1IEP); **PT5** was built by ChemDraw Professional 16. Before the docking simulations, **PT5** and 1IEP preparation included the addition of hydrogens, the assignment of bond order, and assessment of the correct protonation as previously described [46,47]. The MOE 2019.01 software (Chemical Computing Group, Montreal, Canada) was employed for the preparation, interactive docking, visualization and the analysis procedures using its default parameters [48,49,50].

## 4. Conclusions

We herein designed and synthesized benzyl esters of different triterpenes (**PT1**–**PT6**) and the newly substituted benzyl esters of gypsogenin (**GP1**–**GP5**), based on our finding that **1c** (benzyl ester of gypsogenin) has anti-ABL kinase and anti-CML activities [36]. These derivatives and the parent triterpenes were biologically evaluated, and three compounds with good activities, **GP2**, **GP5** and **PT5**, were found. Among them, **PT5** has six times stronger in vitro IC_50_ against ABL kinase compared to **1c**, and stronger activity against CML cell line K562. The stronger interaction of **PT5** with ABL kinase was explained using the difference between the molecular docking by the methyl group in **PT5** on its C4 position and the aldehyde group on the same position of **1c**. The **PT5** was shown to inhibit ERK signaling to induce apoptosis in K562. The other compounds **GP2** and **GP5** have stronger activity against K562 than **PT5**, although their in vitro activity against ABL1 is weaker. It is considered that these two compounds have different mechanisms to induce cell death from inhibition of BCR-ABL. Actually, **GP2** and **GP5** have stronger activity against some cell lines other than K562. Taken together, we succeeded in increasing the activity against ABL kinase and CML cells by derivatization. The strongest inhibitor **PT5** is based on oleanolic acid which is widely spread in many plants. This study would lead to the basis of new anti-CML drugs based on naturally found pentacyclic triterpens.

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
