# Peer review of "Design, Synthesis and Biological Evaluation of Pentacyclic Triterpene Derivatives: Optimization of Anti-ABL Kinase Activity"

_molecules, 2019, doi:10.3390/molecules24193535_

Round 1

Reviewer 1 Report

In this article entitled " Design, Synthesis and Biological Evaluation of Pentacyclic Triterpene Derivatives: Optimization of Anti-ABL Kinase Activity" presented by Prof. Fujita etal., the authors aim to show ABL-kinase inhibition mediated targeted treatment for CML. They have shown that the derivatives exhibit good potency in K-562 cells. Based on selectivity index they evaluated the potent analogues for ABL-kinase binding as a comparison with Imatinib. I would like to recommend the paper for publication in Molecules with minor revision of few following comments.

Many of the analogues tested show moderate to good potency in K-562 cells and analogues GP2, GP5 &PT5 show higher potency/ selectivity similar to Imatinib. When the ABL kinase binding was evaluated GP2, GP5 &PT5 were less potent than Imatinib. Authors may consider arguing about this aspect.

The analogues GP2, GP5 were slightly less potent with respect to kinase binding than PT5. And the authors argue this may be because of different mechanism of action behind inhibition compared to PT5. If 1c and PT5 bind to ABL kinase means GP2 and GP5 may also bind to the kinase though with slight difference in potency due to benzyl group substituion. It will be helpful if more clarity can be given to support this argument.

Regarding greater binding/interaction of PT5 with ABL1 compared to 1c, the authors argue (with the help of molecular docking) that this may be because of the methyl substitution in place of aldehyde. It would be interesting to see whether the 4-cyanobenzyl analogue of PT5 was screened. 

Author Response

Comment 1. Many of the analogues tested show moderate to good potency in K-562 cells and analogues GP2, GP5 &PT5 show higher potency/ selectivity similar to Imatinib. When the ABL kinase binding was evaluated GP2, GP5 &PT5 were less potent than Imatinib. Authors may consider arguing about this aspect.

About GP2 and GP5, we already discussed about the related aspect in the conclusion section of the original manuscript (lines 437-439). We newly added the discussion about PT5 in the text (lines 180-183):

“Although it did not reach to that of imatinib (0.21 µM), similar IC50 values of PT5 and imatinib (~5.5 µM) against K562 cells (Table 1) and these selectivity among various cell lines (Table 2) were observed. PT5 is considered to express function more efficiently in a cell, but the details remain elusive.”.

Comment 2. The analogues GP2, GP5 were slightly less potent with respect to kinase binding than PT5. And the authors argue this may be because of different mechanism of action behind inhibition compared to PT5. If 1c and PT5 bind to ABL kinase means GP2 and GP5 may also bind to the kinase though with slight difference in potency due to benzyl group substituion. It will be helpful if more clarity can be given to support this argument.

GP2 and GP5 have an aldehyde group at C4 that is not in a favorable electrostatic environment when compared to the methyl group of PT5. This explains the pronounced activity of the oleanolic acid-based compound (PT5) over GP compounds. When it comes to 1c, GP2 and GP5 comparison, it seems that electron-withdrawing substitutions on the benzyl moiety slightly improved the activity. This is clearer for the GP5 that has stronger electron-withdrawing p-cyano substitution. 

Comment 3. Regarding greater binding/interaction of PT5 with ABL1 compared to 1c, the authors argue (with the help of molecular docking) that this may be because of the methyl substitution in place of aldehyde. It would be interesting to see whether the 4-cyanobenzyl analogue of PT5 was screened. 

We also think it is curious. However, introduction of p-cyano group to 1c did not show large increase of anti-ABL1 activity (IC50 of GP5: 6.16 mM; 1c: 8.71 mM). Thus, big change can hardly be expected by introduction of p-cyano group to PT5. We think this experiment is not required in this paper.

Furthermore, it was stated that moderate English changes are required. We asked the expert, and English was extensively revised.

Reviewer 2 Report

Ciftci et al. describe new pentacyclic triterpene derivatives with anti-ABL kinase activity. This manuscript has an adequated extension, a validated methodology and reasonable conclusions based on experimental and theoric results.

Some points must be corrected:

In Table 1, the authors must include the IC50 values for PBMC cells for the seven parental compounds (Free triterpenes), and a discussion about the new differences between selectivity and potency of new derivatives relative to parental compounds.

Figure 4, it is presented only one image of Western blot without quantitative analysis. Moreover, I can observe discrepancies in the intensity of bands of phospho-ERK1/2 and cell proliferative IC50 values. No evidence is the relationship between this effect and MTT values reported.

The authors suggest that inhibition of ABL activity, decrease in the p-ERK levels and apoptosis induction are part of the anti-cancer mechanism of new derivatives; however, the manuscript lacks experimental approaches (i.e. rescue, reversions, etc) that show a connection between these events,

Finally, Fig. 4 must include the analysis of total ERK levels (plus the p-ERK levels) and a statistic analysis between the compounds-induced ERK signaling activation.

I suggest for manuscript, a status of “major revision”.

Author Response

Comment 1. In Table 1, the authors must include the IC50 values for PBMC cells for the seven parental compounds (Free triterpenes), and a discussion about the new differences between selectivity and potency of new derivatives relative to parental compounds.

Cytotoxity against PBMC of seven parental compounds was examined, and IC50 values were newly added in Table 1. Some discussion between new derivatives and parental compounds was already written, and new comments were further added to the text (lines 106-109 and lines 111-114):

“Introduction of benzyl ester group into triterpene led to an increase in anticancer activity on K562 except PT3 and PT6, based on GA and UA, respectively. Specially, PT4 has more than 15 times stronger cytotoxicity than its parent triterpene HE.”

“All compounds except HE without cytotoxicity against K562 show selectivity, although each selectivity index (SI) is different. Notably, free triterpenes GP and OA have around IC50 of 13 mM against K562 and SI of more than 7.”.

Comment 2. Figure 4, it is presented only one image of Western blot without quantitative analysis. Moreover, I can observe discrepancies in the intensity of bands of phospho-ERK1/2 and cell proliferative IC50 values. No evidence is the relationship between this effect and MTT values reported.

In Figure 4, intensity of bands was quantitated and shown. To put one representative image of western blot is normal in study of cell signaling. About discrepancy of inhibition of phosphorylated ERK and cytotoxicity against K562, discussion was newly added (lines: 217-218):

“The inhibitory activity against ERK phoshorylation is not inconsistent with cytotoxicity in K562 cells (Table 1). This suggests that there is a pathway other than ERK signaling inhibited by the drugs.”.

Comment 3. The authors suggest that inhibition of ABL activity, decrease in the p-ERK levels and apoptosis induction are part of the anti-cancer mechanism of new derivatives; however, the manuscript lacks experimental approaches (i.e. rescue, reversions, etc) that show a connection between these events.

Main claim in this study is the finding of a new pentacyclic triterpene derivative having anti-ABL and CML activities. The mechanism is shown as suggestion, and the experiments are enough in this paper.

Comment 4. Finally, Fig.4 must include the analysis of total ERK levels (plus the pERK levels) and a statistic analysis between the compounds-induced ERK signaling activation.

Total ERK levels were analyzed, and the imunoblot data and intensity of the bands were newly added to Fig.4. Furthermore, its explanation was added in the text (lines 213-216):

“ERK1/2 and these phosphorylated form in K562-treated with GP2, GP5, or PT5 were examined at 10mM concentration for 6 h (Figure 4). In the presence of compounds, the amount of ERK1/2 moderately decreased. In contrast, the reduction of phosphorylated ERK1/2 was drastic, showing the three compounds exhibited inhibitory effect of phosphorylation of ERK like imatinib.”

In this experiment, one new person (Hiroshi Tateishi) participated in this experiment, thus his name was newly added. To put one representative image of western blot is normal in study of cell signaling.

Reviewer 3 Report

This manuscript will be published on the journal after some additional studies.

According to the table1、 2、 and 3, some new triterpenes will be synthesized from oleanolic acid with electron-withdrawing benzyl ester maybe show more high activities.

Author Response

Comment. According to the table1, 2 and 3, some new triterpenes will be synthesized from oleanolic acid with electron-withdrawing benzyl ester maybe show more high activities.

We also think it is curious. However, introduction of electron withdrawing group to 1c did not show large increase of anti-ABL1 activity (IC50 of GP2: 7.19 microM: GP5: 6.16 microM; 1c: 8.71 microM). Thus, big change can hardly be expected by introduction of electron withdrawing group to PT5. We think this experiment is not required in this paper.

Furthermore, it was stated that moderate English changes are required. We asked the expert, and English was extensively revised.

Reviewer 4 Report

The manuscript describes synthesis of 11 compounds by esterification of the carboxylic functional group of natural triterpenes by a standard protocol. The products were evaluated for their antiproliferative and anti-ABL kinase activities.  Even though the design of the compounds is unclear, the products demonstrated some biological activity.

In general, the manuscript is reasonably well written, but there are some important issues making credibility of this manuscript questionable. These issues are related to the structural characterisation of the prepared compounds: 

1) There is no evidence of compounds' purity (no EA, qNMR, or HPLC data). Therefore, biological assay results might be affected.

2) NMR spectra description in the experimental section is totally confusing with the same regions reported more than once for the same compound. For example, line 226:

"2.26 - 0.9 (m 20H)" and then in the same and the following line overlapping values: "1.07 (s, 3H), 1.00 (s, 3H), 0.93 (d, J = 6.3 Hz, 3H)".

The same problem is repeated for each compound! 

Moreover, an attempt to clarify this issue using Supplementary Information resulted in an even greater confusion. In the spectra, many signals are not integrated or their integrations do no correspond to the reported values. There is no a single 1H NMR spectra properly processed!

All NMR spectra should be properly processed, analysed and an accurate signal assignments should be done and presented in the revised manuscript for the further reviewing. In the present state their assessment is impossible. 

Author Response

Comment 1. There is no evidence of compounds' purity (no EA, qNMR, or HPLC data). Therefore, biological assay results might be affected

In all compounds, TLC, H1 and C13 NMR spectra showed no considerable suspect peak that can be impurity. The H1 and C13 NMR spectra are shown in Supplementary information.

Comment 2. NMR spectra description in the experimental section is totally confusing with the same regions reported more than once for the same compound. For example, line 226:

"2.26 - 0.9 (m 20H)" and then in the same and the following line overlapping values: "1.07 (s, 3H), 1.00 (s, 3H), 0.93 (d, J = 6.3 Hz, 3H)".

The same problem is repeated for each compound! 

Moreover, an attempt to clarify this issue using Supplementary Information resulted in an even greater confusion. In the spectra, many signals are not integrated or their integrations do no correspond to the reported values. There is no a single 1H NMR spectra properly processed!

All NMR spectra should be properly processed, analysed and an accurate signal assignments should be done and presented in the revised manuscript for the further reviewing. In the present state their assessment is impossible. 

We retried the NMR interpretation, confirmed the integration accuracy, and avoided any overlapping. Now, the total number of integrals matches with the total number of protons for each compound. Furthermore, we successfully assigned the benzyl moiety protons for each compound to ensure the conversion of the free pentacyclic triterpenic acids into the corresponding benzyl esters.

Furthermore, it was stated that minor English spell check is required. We asked the expert, and English was extensively revised.

Round 2

Reviewer 2 Report

I consider that the new version of this manuscript is a candidate to be published in Molecules Journal.

Author Response

Comment. I consider that the new version of this manuscript is a candidate to be published in Molecules Journal.

Thank you very much for your positive comment.

Reviewer 4 Report

Unfortunately, the earlier queries have not been fully addressed by authors. There is a great improvement in the presentation of spectra in the Supplementary Materials and some improvements in the data presentation in the manuscript. However, some issues still remain. Event though a partial (for the benzyl group) assignment of signals 1H NMR spectra have been done and there are no obvious overlaps in the signals reporting, 13C NMR spectra still require additional attention. The current way of reporting 13C NMR spectra is misleading and does not give an opportunity for a skilled person to attempt a signal assignment. All signals are listed in the order of appearance as if each of them is a singlet. However, for fluorinated compounds GP2, GP3, and GP4 the aromatic ring (and CF3 for GP2 and GP3) signals of carbon atoms are obviously split by fluorine. These signals should be reported as multiplets corresponding to their spitting with C-F coupling constants included.

In terms of English, I cannot see any significant improvement and suggest seeking help from an English speaking expert / editor. 

Author Response

Comment 1. Unfortunately, the earlier queries have not been fully addressed by authors. There is a great improvement in the presentation of spectra in the Supplementary Materials and some improvements in the data presentation in the manuscript. However, some issues still remain. Event though a partial (for the benzyl group) assignment of signals 1H NMR spectra have been done and there are no obvious overlaps in the signals reporting, 13C NMR spectra still require additional attention. The current way of reporting 13C NMR spectra is misleading and does not give an opportunity for a skilled person to attempt a signal assignment. All signals are listed in the order of appearance as if each of them is a singlet. However, for fluorinated compounds GP2, GP3, and GP4 the aromatic ring (and CF3 for GP2 and GP3) signals of carbon atoms are obviously split by fluorine. These signals should be reported as multiplets corresponding to their spitting with C-F coupling constants included.

The C-F splitting is well-noticed in 13C spectra. We picked the splitted peaks at the 13C charts in the supplementary information and re-assigned 13C spectra of compounds GP2, GP3 and GP4 in the main text as recommended.

Comment 2. In terms of English, I cannot see any significant improvement and suggest seeking help from an English speaking expert / editor. 

English was further improved by the expert.